# *Meta*-Topolin as an Effective Benzyladenine Derivative to Improve the Multiplication Rate and Quality of In Vitro Axillary Shoots of Húsvéti Rozmaring Apple Scion

**DOI:** 10.3390/plants13111568

**Published:** 2024-06-06

**Authors:** Neama Abdalla, Judit Dobránszki

**Affiliations:** 1Plant Biotechnology Department, Biotechnology Research Institute, National Research Centre, 33 El Buhouth St., Dokki, Giza 12622, Egypt; 2Centre for Agricultural Genomics and Biotechnology, Faculty of Agriculture and Food Sciences and Environmental Management, University of Debrecen, 4400 Nyíregyháza, Hungary

**Keywords:** apple, Rosaceae, in vitro, shoot multiplication, cytokinin

## Abstract

In vitro mass propagation of apple plants plays an important role in the rapid multiplication of genetically uniform, disease-free scions and rootstocks with desired traits. Successful micropropagation of apple using axillary shoot cultures is influenced by several factors, the most critical of which is the cytokinin included in the culture medium. The impact of medium composition from single added cytokinins on shoot proliferation of apple scion Húsvéti rozmaring cultured on agar-agar gelled Murashige and Skoog medium fortified with indole butyric acid and gibberellic acid was investigated. The optimum concentration for efficient shoot multiplication differs according to the type of cytokinin. The highest significant multiplication rate (5.40 shoots/explant) was achieved using 2.0 μM thidiazuron while the longest shoots (1.80 cm) were observed on the medium containing benzyladenine at a concentration of 2.0 μM. However, application of either thidiazuron or benzyladenine as cytokinin source in the medium resulted in shoots of low quality, such as stunted and thickened shoots with small leaves. In the case of benzyladenine riboside, the 8 μM concentration was the most effective in increasing the multiplication rate (4.76 shoots/explant) but caused thickened stem development with tiny leaves. In the present study, *meta*-topolin was shown to be the most effective cytokinin that could be applied to induce sufficient multiplication (3.28 shoots/explant) and high-quality shoots along with shoot lengths of 1.46 cm when it was applied at concentrations of 4 μM. However, kinetin was the least active cytokinin; it practically did not induce the development of new shoots. The superior cytokinin for in vitro axillary shoot development of apple scion Húsvéti rozmaring with high-quality shoots was the *meta*-topolin, but it may be different depending on the variety/genotype under study.

## 1. Introduction

Apple (*Malus × domestica* Borkh), a member of family Rosaceae, is one of the most economically significant, highly consumed, and popularly grown fruit crops with nutritive and medicinal importance needed for diet therapy and adjuvant treatment [1,2]. It is widely planted in temperate climate regions of the world [3,4]. Due to the biological reproductive characteristics of apple trees, such as prolonged juvenility, frequent self-incompatibility, and high level of heterozygosity [5], apple trees are vegetatively propagated by conventional methods, such as grafting apple scion cultivar onto clonal rootstocks [6], or layering, budding, and cutting [7], to maintain a consistent genetic background of the desired cultivars [8]. However, these traditional methods of propagation cannot guarantee obtaining healthy, pathogen-free plant materials. They rely on the cultivation season and lead to lower multiplication rates that reflect negatively on apple production [9].

Since micropropagation of apple was reported for the first time more than five decades ago [10], tissue culture has been broadly used for providing multiple clones of apple scions and rootstocks [1] via many techniques (i.e., by shoot culture, meristem culture, and stem node culture, or by adventitious somatic embryos or adventitious shoots, micrografting, or thin cell layer from leaves [9]). Micropropagation of apple is employed to produce genetically uniform disease-free scions and rootstocks for commercial production in large scale [11]. Micropropagation of apple rootstocks could be used to overcome the limitations of traditional methods [3], and offer a promising tool for efficient, cost-effective, and rapid clonal multiplication protocol of apples and other fruit tree plants [12,13]. Further, micropropagation permits fast propagation of new varieties or lines for apple breeders [14]. Also, it is applied to induce in vitro polyploidization of apple genotypes to reduce the time required for breeding programs and to enhance the resistance to biotic or abiotic stresses [15]. Moreover, it is essential for regeneration of transgenic lines [16], and for efficient transformation protocols [17,18]. In addition, it could be utilized to produce virus free rootstocks by in vitro thermotherapy-, cryotherapy-, and/or chemotherapy-based techniques used for virus eradication [19,20]. In vitro tissue culture of apple can be utilized to screen the biotic and abiotic stress tolerance in apple rootstocks and scions [21,22]. It is also useful in cryopreservation of germplasm or genetic materials and production of synthetic seeds [1].

In vitro shoot proliferation of apple has been reported to be genotype-dependent and influenced by various factors, such as type and size of explant, age of mother plant, and culture conditions (including photoperiod, light intensity, and quality) [3]. However, one of the most essential factors affecting the process of shoot regeneration is the type and concentration of cytokinin supplemented in the medium [12]. Shoot regeneration via organogenesis, either axillary or adventitious, has been described for different apple rootstocks and scions from different explants, like axillary buds and shoot apices or leaves [2,3,23,24,25,26,27,28,29,30,31,32,33,34,35,36], leaf transverse thin cell layers [37,38], meristem culture [39], single node cutting [40,41,42], nodal segments [43], and stem explants [33].

Cytokinins (CKs) are a chemically unique group of plant growth promoters, displaying a wide range of actions in regulating plant growth, morphology, and development [44]. The significance of CKs in clonal micropropagation of fruit tree species has been recognized [45]. They play an essential role in in vitro shoot organogenesis as they stimulate cell division and initiate growth and proliferation of buds and shoots in tissue cultured plants [46,47]. Thidiazuron (N-phenyl-N-1,2,3 thiadiazol-5-ylurea, TDZ) and benzyladenine (BA) are the most frequently used cytokinins in apple shoot regeneration protocols, but their potential is mainly dependent on genotype and other in vitro factors [3,28]. BA has been the preferred cytokinin for apple shoot multiplication; however, other analogs of benzyladenine, such as benzyladenine-9-riboside (BAR), or hydroxylated BA analogues, such as *meta*-topolin, *m*T [6-(3-hydroxybenzylamino) purine)], could enhance shoot proliferation [3], and could be applied as an alternative cytokinin to avoid the undesirable side effects induced by BA or TDZ during shoot regeneration (hyperhydricity, rooting difficulties, stunted shoots, toxicity, and callus formation). *m*T has been proven to be one of the most effective aromatic cytokinins applied in tissue culture of some plant species as well as in apple micropropagation [3]. Other cytokinins, such as kinetin (6-furfurylaminopurine, KIN), have also been tested; however, kinetin was recorded to have less activity when applied alone and its effect was genotype-dependent [3,28].

The optimal concentration of cytokinins needed for apple shoot regeneration varies by genotype and it can also depend on the type and size of the explant [3,28]. The success of a micropropagation protocol based on tissue-culture techniques with the aim of commercial production at a large scale largely depends on the mode and rate of shoot multiplication [48]. Húsvéti rozmaring (HR), or Entz rozmaring, apple scion is considered one of the most popular and traditional old apple varieties of gene-preserving and gene-resource importance, which is well known and widespread in Hungary. The exact origin of this variety is not clear, but it may have originated in the Great Hungarian Plain area that lies between the Tisza and Danube rivers and is known for its sandy soils. It can be used as a source of resistance in breeding programs of apple plants due to its tolerance against different biotic and abiotic stresses. The resistance of this variety makes it a good candidate for organic cultivation. It is a diploid, self-fertile and late-ripening cultivar. It shows excellent post-harvest ability; its fruits are usually harvested in October and can be stored at 10 °C until about Easter time, at the end of March. Its fruits are yellowish-green with a red blush color at ripening [24,49].

The current investigation aimed to examine the single effect of different cytokinins (TDZ, BA, BAR, *m*T, and KIN) added separately at different concentrations (0, 2, 4, 6, and 8 μM) to Murashige and Skoog (MS) medium containing 0.49 μM indole butyric acid (IBA) and 0.58 μM gibberellic acid (GA_3_) on in vitro shoot multiplication of apple scion Húsvéti rozmaring.

## 2. Results

### 2.1. Effect of Different Cytokinins on In Vitro Shoot Development from Axillary Buds

The main effect of each individual factor under study (cytokinin type and concentration) on shoot fresh weight, multiplication rate, and shoot length of apple scion HR from in vitro shoot cultures was evaluated. Different CKs added separately at different concentrations to MS medium contained IBA as auxin at 0.49 μM and gibberellin (GA_3_) at 0.58 μM. In general, TDZ was the optimal CK for shoot fresh weight; it was significantly superior over the other CKs under investigation, while the concentration of 8 μM of CK recorded the highest significant value of shoot fresh weight regardless of the type of cytokinin. The data presented in Table 1 show significant differences among CK treatments at *p* ≤ 0.05. The first level of TDZ (2 μM) recorded the highest value of shoot fresh weight, then a decrease was noticed in fresh weight at the second level (4 μM) followed by an increase at the third to the fourth levels (6 and 8 μM, respectively). BA increased shoot fresh weight from the first to the second level, then a slight decrease happened at the third level, then an increase again and the maximal fresh weight (9070.2 mg) was recorded for BA at 8 μM. There was a positive correlation between shoot fresh weight and BAR concentrations where, by increasing the applied level of BAR, shoot fresh weight increased; then the maximum value (8703.3 mg) of fresh weight was measured for BAR at 8 μM. *m*T reached its maximum recorded value (9258.5 mg) at 4 μM. The maximum fresh weight (9670.1 mg) was achieved on MS medium containing 0.49 μM IBA + 0.58 μM GA_3_ and supplemented with 2 μM TDZ, followed by 8 μM TDZ, 4 μM *m*T, and 8 μM BA, without significant differences among them. The control medium (CK-free MS) resulted in the lowest (2768.6 mg) shoot fresh weight. In response to KIN treatments, the biomass of the original shoots increased, but without further new shoot development therefore it was not presented in the Table 1. The most effective CKs for shoot biomass in terms of shoot fresh weight were determined as TDZ, BA, *m*T, and BAR, in descending order, even if without significant differences between them.

Concerning the multiplication rate, the main effect of cytokinin type was recorded to be significant. TDZ was the most efficient CK for enhancing number of shoots/explant and CK levels of 4, 6, and 8 μM significantly recorded the highest values of shoot number concerning the main effect of the concentration. The data in Table 2 indicate that MS medium containing 0.49 μM IBA + 0.58 μM GA_3_ and fortified with 2 μM TDZ produced the maximum number of shoot (5.40 shoots/explant), and it was not significantly different from all the other treatments of TDZ except control (zero μM TDZ), most of BA treatments (4, 6, and 8 μM), and 8 μM BAR. On the other hand, KIN treatments recorded the same non-significant response that was noticed for the control. The worst responses were obtained with all concentrations of KIN as well as the control (CK-free); in those mediums, the original shoots developed further, practically without developing new shoots. It was observed that TDZ at 6 μM, and BA and BAR at 8 μM had an equal effect on shoot number without significant differences to each other.

Unlike shoot fresh weight and shoot number where TDZ was the best added CK, BA then BAR were the optimal CKs that significantly increased shoot length, followed by *m*T, compared to TDZ, control, and KIN. On the other hand, TDZ recorded a weak effect on shoot length, as shown in Table 3. The longest shoot (1.80 cm) was produced with 2 μM BA (supplemented in MS medium containing 0.49 μM IBA + 0.58 μM GA_3_). However, it was not significantly different from some other concentrations of BA (4 and 8 μM), some concentrations of BAR (2, 4, and 6 μM), and *m*T at 4 and 8 μM. The control, CK-free medium, resulted in the shortest shoot (0.15 cm). This was not significantly different from TDZ at 4 μM, *m*T at 2 μM, and all concentrations of KIN. It could be proven that TDZ at 2 μM was the best for enhancing shoot multiplication rate of HR apple while BA applied at the same concentration was the most beneficial for shoot elongation. A negative correlation between shoot number and shoot length was detected especially when TDZ was applied. At a concentration optimal for shoot multiplication, an inhibition of shoot length was noticed.

### 2.2. Effect of Different Cytokinins on Morphology of In Vitro Shoots

In Figure 1, the developed in vitro shoots from optimal concentrations of each CK-type are presented. On the control, CK-free medium (Figure 1e), the shoot explants developed further, just like on KIN-containing medium (Figure 1f). Development of new shoots was not a trait on those media. Leaves on CK-free medium were large and well expanded. Similar morphology could be observed on media supplemented with KIN, but by the end of the subculture leaves showed some ageing and started to discolor. On the 8 µM BA-containing medium, the developed shoots had long but thickened stems with small leaves (Figure 1a). Stems were rigid, and difficult to cut. Similarly, on 8 µM BAR-containing (Figure 1c) medium new shoots had thickened stem with tiny leaves. Shoots developed on the medium containing 2 µM TDZ (Figure 1b) were stunted, of a dwarf nature that make its separation into explants and the establishment of the next subculture difficult. In addition, leaves on this medium were pale green, in contrast to the deeper green leaves developed on other media, mainly the optimal, *m*T-containing medium (Figure 1d). Among CKs tested, new shoots developed on 4 µM *m*T showed the best quality with large, well-expanded leaves (Figure 1d).

## 3. Discussion

Shoot proliferation and multiplication during micropropagation process is affected by the type and level of PGRs applied in culture media; CKs especially have governing effects due to their significant functions in cell division, organogenesis, and suppressing the apical dominance, thus enhancing shoot formation from lateral buds [50,51,52]. CKs are widely added in shoot cultures to regulate the in vitro growth and development of newly regenerated shoots [44]. However, the choice of cytokinin remains the most crucial factor to the success or failure of any in vitro propagation protocol [53].

Extensive laboratorial research work has been carried out to optimize tissue culture protocols with considerable emphasis on the type of CK and their use in improvement of apple plants [45]. These protocols are developed and standardized based on quantitative and qualitative evaluation of growth and developmental parameters such as shoot multiplication rate, rooting, and acclimatization efficiency [54]. In this context, shoot multiplication, rooting, regeneration, and transformation methodologies have been developed in apple plants [1]. Shoots branching from lateral meristems of apple depends on the initiation and activation of axillary buds, which are hormonally regulated by cytokinins as the major PGR in culture medium [3]. An earlier report on single cytokinin-effect on shoot multiplication of different apple scions (‘Prima’, ‘Galaxy’, and ‘Jonagold’) was published [23]. It was found that the rate of newly developed shoots depended on the type and concentration of cytokinin as well as on genotype.

Previous reports discussed in vitro shoot regeneration and development of a large number of apple scion and/or rootstock [3,9,12,23,25,30,32,34,36,37,45,55,56,57,58], or wild apple varieties used for genetic conservation [33]; however, few of them focused on the Húsvéti rozmaring apple scion [24]. Most studies have reviewed the response of in vitro shoot cultures of apple from different cultivars to various concentrations of BA ranging from 1 to 10 μM. The optimum level of BA that produced the maximum number of shoots varied according to the cultivar. It improved shoot multiplication of eight different apple cultivars (‘Cacharela’, ‘Camoesa’, ‘Repinaldo’, ‘Tres en Cunca’, ‘Gravillan’, ‘Ollo Mouro’, ‘Jose Antonio’, ‘Principe Grande’) when applied at 1.0 mg L^−1^ (4.4 μM) to apical bud cultures [12]. Therefore, it was recommended to use 4.4 μM BA for shoot proliferation for most apple cultivars. This concentration allows longer shoots to develop, but shoot number was suboptimal [3]. TDZ stimulated shoot multiplication of ‘Gala’ apple when added to the medium in a concentration ranging between 0.1 to 10 μM [3]. However, shoots were shorter and leaves were smaller on TDZ-containing medium than on medium with BA. Moreover, formation of large shoot clumps was observed on TDZ-supplemented medium and some of these shoots looked adventitious, which may be unfavorable for producing genetically uniform, true-to-type, or identical plants [3]. TDZ at 2.27 μM was optimum for adventitious shoot multiplication (11 shoots/explant) from leaves of ‘Royal Gala’ apple [56,57]. These earlier findings were in agreement with our recent obtained results, even if the shoot development was axillary; 2 μM TDZ was the optimal concentration for maximum shoot number (5.40), however short shoots (0.51 cm) were produced. This could suggest supplying culture medium with 4 μM BA to obtain a proper number (4.20) of elongated shoots of 1.51 cm length from the shoot culture of HR apple.

To maximize the efficiency of in vitro shoot proliferation and avert the bad side effects of BA and TDZ such as difficulties rooting following in vitro shoot propagation or toxicity after several subcultures. different derivates of BA were also examined during in vitro shoot multiplication of apple. A better multiplication rate was achieved in ‘MM.106’ and in ‘JTE-H’ apple rootstock when BA was substituted by BAR. It was proved that BAR increased the rate of multiplication in both rootstocks when compared to BA [59]. In contrast and according to the current study, BA seemed to be more effective than BAR in enhancing shoot multiplication rate of HR; however, the effects of BAR and BA were not significantly different at 8 μM. This result confirmed that the response of apple to the type and concentration of CKs applied during in vitro shoot multiplication mainly depends on the genotype or cultivar under investigation.

Researchers are continually searching for newly discovered and potent CKs. The group of topolins in general and *meta*-topolin (*m*T) particularly are products of such efforts. Since the discovery of topolins as aromatic cytokinins with a natural origin, they have recently begun being used as true substitutes to the commonly used CKs such as benzyladenine, zeatin, and kinetin in plant tissue culture medium. *m*T has been applied for culture establishment, protocol optimization, and for alleviating different in vitro induced physiological abnormalities and disorders in many plant species [54]. It could prevent shoot-tip necrosis, and help evade the effects of hyperhydricity. It has tremendous potential in in vitro shoot regeneration by delaying senescence, which ultimately facilitates multiple shoot induction, proliferation, and regeneration, and increases shoot length [52,60]. *m*T has proven to be an effective cytokinin for inducing multiple shoot proliferation in a number of plant species via direct and indirect organogenesis [61]. Nevertheless, adverse and undesirable responses to topolins applied for shoot regeneration have also been detected in certain plant species [62]. A lower multiplication rate had been reported in Rosa hybrids for topolin than that recorded for BA treatment. The same results were documented for *m*T in shoot multiplication of HR apple compared to BA and TDZ. This confirmed that there is no exact pattern in the responses of plant species to different CKs because it is genotype-dependent [54], and it is affected by the CK concentration used as well as medium type [56]. These vital factors should be taken into account while optimizing PTC protocols.

The post-effects of *m*T on rooting ability of micropropagated shoots were examined earlier. It influenced positively shoot rooting and acclimatization capacity at ex vitro conditions of some apple cultivars when it was added at 8 μM in the last multiplication subculture instead of BA [63]. It induced 95% rooting of the in vitro shoots that regenerated from the leaves of ’Red Fuji’ apple when added at 1 mg L^−1^ (≈4.15 μM) to the proliferation medium [64]. Moreover, it stimulated shoot regeneration from shoot explants of ‘Royal Gala’ apple at 2.1 μM [25], and increased the number of shoots per leaf segment of RG apple (up to 15.1) at 1.5 mg L^−1^ (6.2 μM; [55]) and Jonagold apple at 5.0 mg L^−1^ (20.7 μM; [65]). These results agree with our recent findings on the HR cultivar, where *m*T was able to support shoot multiplication and the growth of shoots. *m*T at 4 μM gave the maximum results (3.28) recorded for shoot number. These findings confirmed that the optimal concentration of *m*T for shoot multiplication of apple depended not only on the examined genotype but also on the type of explant used, as reported by Gantait and Mitra [52]. Other aspects of *m*T, apart from effective induction and multiplication of shoots, are the increase in shoot length, fresh weight, and photosynthetic capacity of in vitro-derived plants [44,66]. Similarly, and according to the recent investigation, *m*T increased shoot fresh weight and shoot length at 4 and 8 μM, respectively, and resulted in high quality in vitro developed shoots according to the morphology of newly developed shoots.

The quality, morphology and the physiological status of microshoots are of great importance in their subsequent development (adventitious or axillary shoot regeneration, rooting) and acclimatization when transferring the micropropagated plantlets from in vitro to ex vitro conditions at which they will be autotrophic [3]. Applying *m*T for in vitro propagation could be useful for inducing efficient axillary shoot proliferation, which is the most frequently used technique for commercial mass propagation, with a high multiplication rate and healthy shoots. *m*T-treated plantlets had the best acclimatization ability, which could be due to the high shoot quality at the multiplication stage. Moreover, *m*T enhanced chloroplast differentiation, lessened chlorophyll degradation, modified antioxidant enzyme activities, and, as a result, improved rooting and increased the acclimatization capacity [67]. *m*T positively affected the function of the photosynthetic apparatus and the pigment content of in vitro leaves of Royal Gala apple scion. The maximum chlorophyll a/b ratio that could support increasing the survival of the plantlets at acclimatization was observed at 6.0 μM *m*T [44]. The absence of malformations and somaclonal variation are further advantages of *m*T [67]. *m*T at 1 mg L^−1^ (≈4.15 μM) increased the survival percentage and decreased hyperhydricity of plantlets regenerated from meristems cultures of ‘Golden Delicious’ apple [68]. It enhanced the quality of *Prunus domestica* L. and *Prunus insititia* × domestica in vitro shoots at 2.1 μM [69], and elongated shoots with an increased number of internodes and large leaves of *Prunus insititia* × *domestica* ‘Ferdor’ and *P. domestica* ‘Torinel’ [70]. An optimum multiplication rate and high-quality shoots of in vitro grown pear rootstock OHF-333 were obtained on MS medium containing 6–9 μM *m*T, leaf gas exchange was improved, and phenol content was decreased [71].

The multiplication rate of in vitro shoot cultures, as influenced by repeated subculturing, varies according to plant species and even varieties. A decline in multiplication potential after repeated subcultures on a medium of unchanged hormonal composition was reported in some cultivars of Rosaceae. Shoot proliferation capacity of cherry, plum, and pear rootstocks decreased over frequented subculturing in all examined genotypes [72]. Thus, the current investigation focused on the in vitro shoot multiplication of Húsvéti rozmaring apple scion for one subculture (four weeks). It is clear that the majority of cytokinin types (BA, TDZ, KIN, and even BAR) can be excluded as potential cytokinins that can be used for the shoot multiplication of this scion because their effects were non-beneficial already in the first subculture. Therefore, in the next phase of our research, we intend to study the effects of subcultures, and the potential after-effects of mT on the subsequent rooting and acclimatization.

## 4. Materials and Methods

This study was conducted at Plant Biotechnological Laboratory, Centre for Agricultural Genomics and Biotechnology, Faculty of the Agricultural and Food Science and Environmental Management, University of Debrecen, Nyíregyháza, Hungary.

### 4.1. Plant Materials

Pre-established in vitro shoot cultures of apple (*Malus* × *domestica* Borkh), scion cultivar Húsvéti rozmaring (HR), of four weeks old were used as a source of explants (shoots) in the current investigation.

### 4.2. In Vitro Propagation of Plant Materials

The microshoots of HR apple were separated from its in vitro mother shoot, then the lateral leaves were removed, so the separated shoots contained the shoot tip and the axillary buds. These shoots were cultured horizontally on MS [73] medium (100 mL L^−1^ MS macro, 1 mL L^−1^ MS micro, 1 mL L^−1^ MS vitamin, 1 mL L^−1^ Humus), and fortified with 0.5 mg L^−1^ BA, 0.1 mg L^−1^ IBA (0.49 μM), and 0.2 mg L^−1^ GA_3_ (0.58 μM), containing 3% (*w*/*v*) sucrose and 0.7% (*w*/*v*) agar-agar. The pH of the medium was adjusted to 5.8 before autoclaving, then the medium was autoclaved for 20 min at 121 °C and 10^5^ Pa. The explants were cultured for 4 weeks in cylindrical glass jars of 370 mL capacity containing 70 mL medium for axillary shoot induction. Cultures were incubated at 23 ± 2 °C, under a 16/8 photoperiod (1:1 mix of daylight and warm white fluorescent lamps in a vertical position), at light intensity of 106 µmol s^−1^ m^−2^.

### 4.3. Transfer of the Shoots to Cytokinin-Free Medium

The multiplicated shoots of HR apple were separated and prepared as described above. Shoot explants were cultured on cytokinin-free MS medium and supplemented with 3% (*w*/*v*) sucrose and 0.7% (*w*/*v*) agar-agar. The pH of the medium was adjusted to 5.8 before autoclaving, then the medium was autoclaved for 20 min at 121 °C and 10^5^ Pa. The explants were cultured in cylindrical glass jar of 370 mL capacity containing 70 mL medium. Cultures were incubated at 23 ± 2 °C, under a 16/8 photoperiod (1:1 mix of daylight and warm white fluorescent lamps in a vertical position), at light intensity of 106 µmol s^−1^ m^−2^. After 4 weeks on cytokinin-free medium, the shoots were separated as described above, then transferred to the cytokinin treatments where different cytokinins (i.e., TDZ, BA, BAR, *m*T, and KIN) were added individually at different concentrations (i.e., 0, 2, 4, 6, and 8 μM) to MS medium supplemented with 3% sucrose and 0.7% agar-agar (5 cytokinins × 5 concentrations = 25 treatments). The pH of the medium was adjusted to 5.8 before autoclaving, then the medium was autoclaved for 20 min at 121 °C and 10^5^ Pa. The same glass jars were used as for maintenance and multiplication of cultures, and the in vitro environmental conditions were the same, as well, as applied before. After 4 weeks on cytokinin-containing media, the fresh weight of microshoots (mg), multiplication rate (number of newly developed microshoots/explant), and length of microshoots (cm) were measured from five jars where each jar contained 5 shoot explants (1.5–2 cm) at the beginning of the experiments.

### 4.4. Statistical Analyses

SPSS for Windows software (SPSS^®^, version 21.0) was applied for analysing the data. Univariate Analysis of Variance followed by Tukey’s test (at *p* < 0.05) was applied to assess main effects of different factors, i.e., the type and concentration of cytokinin, as well as the interactions between individual variables. One-way ANOVA followed by Tukey’s test (at *p* < 0.05) was used to analyse the effects of different treatments.

## 5. Conclusions

Tissue culture has been widely employed for obtaining multiple clones (micropropagation) of apple rootstocks and scions. Furthermore, it includes biotechnology-based tools for apple crop improvement. The success of micropropagation protocol depends on various factors such as explant type, collection time, age of donor plant, genotype, carbohydrate source, components of culture medium, plant growth regulator, pH, and culture conditions. The choice of CK is one of the most critical factors in developing an effective PTC protocol. CKs are often the main determinants of successful plant regeneration. It should, however, be noted that plant species respond differently to various cytokines and their derivatives. Thus, to optimize PTC protocols, strict selection of CKs is essential. Despite the medium shoot multiplication rate noticed for *m*T in shoot cultures of HR apple, and to avoid the harmful disorders resulting from the application of TDZ and BA during in vitro shoot regeneration of apple, it could be recommended to use *m*T as an alternative CK for multiplication of in vitro shoots of high quality from the axillary bud of HR apple at 4 μM. Also, it could be applied for shoot elongation to increase shoot length at a concentration of 8 μM. Further examinations are necessary to determine the potential after-effects of the different CKs in subsequent shoot multiplications and in vitro rooting and acclimatization in the case of Húsvéti rozmaring scion.

## Figures and Tables

**Figure 1 plants-13-01568-f001:**
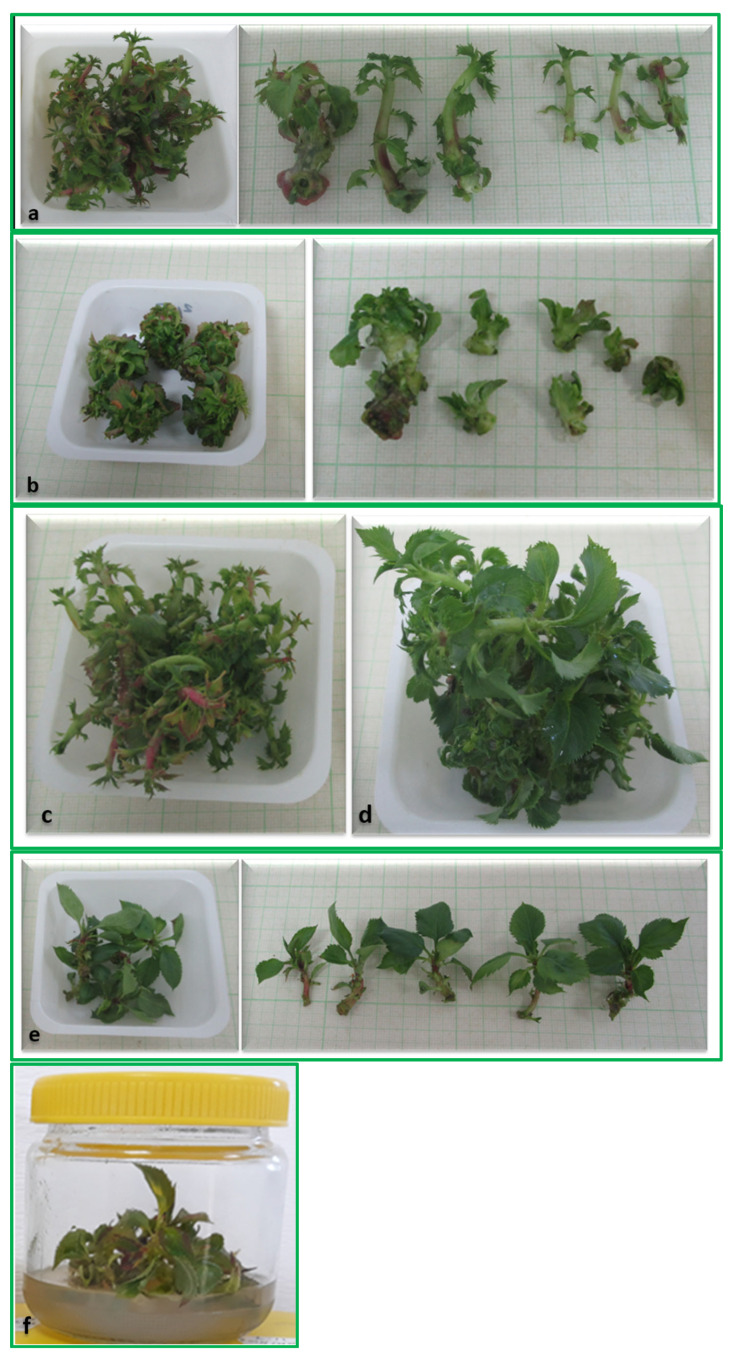
In vitro axillary shoot development of Húsvéti rozmaring apple scion in response to the various CK contents of the medium. Optimal concentration of each CK on shoot development is shown: (**a**) 8 μM BA; (**b**) 2 μM TDZ; (**c**) 8 μM BAR; (**d**) 4 μM *m*T; (**e**) CK-free medium; (**f**) 2 μM KIN as the worst CK.

**Table 1 plants-13-01568-t001:** Effect of different concentrations of TDZ, BA, BAR, and *m*T supplemented in MS medium (containing 0.49 μM IBA + 0.658 μM GA_3_) on the fresh weight of new shoots developed from in vitro shoot cultures of apple scion Húsvéti rozmaring after four weeks of culture.

Shoot Fresh Weight (mg)					
Main effect of different factors					
Cytokinin *** Concentration *** Cytokinin × Concentration interaction ***					
Cytokinins					
TDZ		BA	BAR	*m*T	
8647.7 ± 337.9 a		7285.8 ± 439.8 b	6595.8 ± 534.0 b	6339.1 ± 641.9 b	
Concentrations (μM)					
Ø		2	4	6	8
2768.6 ± 191.8 d		5601.5 ± 487.2 c	7113.1 ± 415.1 ab	6309.8 ± 459.8 bc	7875.5 ± 410.9 a
Treatments					
μM	TDZ	BA	BAR	*m*T	
Ø	2768.6 ±191.8 g				
2	9670.1 ± 721.6 a	4965.4 ± 370.8 c	3876.0 ± 545.8 fg	4805.6 ± 378.5 defg	
4	7176.1 ± 331.3 abcde	7635.1 ± 846.5 abc	6171.2 ± 698.2 bcdef	9258.5 ± 1246.8 a	
6	8385.2 ± 591.5 ab	7472.4 ± 611.2 abcd	7632.8 ± 988.1 abc	3716.4 ± 228.9 fg	
8	9359.3 ± 493.3 a	9070.2 ± 515.9 a	8703.3 ± 618.8 ab	7269.4 ± 903.5 abcde	

Values followed by the same letter in the same column are not significantly different at *p* ≤ 0.05 level, according to Tukey’s multiple range test. Ø (CK-free MS medium) served as a control. *** means *p* ≤ 0.001.

**Table 2 plants-13-01568-t002:** Effect of different concentrations of TDZ, BA, BAR, *m*T, and KIN supplemented in MS medium (containing 0.49 μM IBA + 0.58 μM GA_3_) on shoot multiplication rate from in vitro shoot cultures of apple scion Húsvéti rozmaring after four weeks of culture.

Number of Newly Developed Microshoots/Explant					
Main effect of different factors					
Cytokinin *** Concentration *** Cytokinin × Concentration interaction ***					
Cytokinins					
TDZ		BA	BAR	*m*T	KIN
4.76 ± 0.154 a		3.96 ± 0.177 b	3.47 ± 0.205 c	1.73 ± 0.177 d	0.02 ± 0.014 e
Concentrations (μM)					
Ø		2	4	6	8
0.26 ± 0.066 c		2.29 ± 0.208 b	3.05 ± 0.189 a	2.89 ± 0.218 a	3.11 ± 0.221 a
Treatments					
μM	TDZ	BA	BAR	*m*T	KIN
Ø	0.26 ± 0.066 h				
2	5.40 ± 0.321 a	2.84 ± 0.263 ef	2.16 ± 0.309 fg	0.55 ± 0.198 h	0.04 ± 0.042 h
4	4.28 ± 0.286 abcd	4.20 ± 0.374 abcd	3.12 ± 0.318 def	3.28 ± 0.319 cdef	0.00 ± 0.000 h
6	4.92 ± 0.282 ab	4.16 ± 0.340 abcd	3.84 ± 0.411 bcde	1.10 ± 0.275 gh	0.04 ± 0.042 h
8	4.44 ± 0.311 abc	4.64 ± 0.346 ab	4.76 ± 0.421 ab	1.67 ± 0.299 g	0.00 ± 0.000 h

Values followed by the same letter in the same column are not significantly different at *p* ≤ 0.05 level, according to Tukey’s multiple range test. Ø (CK-free MS medium) served as a control. *** means *p* ≤ 0.001.

**Table 3 plants-13-01568-t003:** Effect of different concentrations of TDZ, BA, BAR, *m*T, and KIN supplemented in MS medium (containing 0.49 μM IBA + 0.58 μM GA_3_) on shoot length from in vitro shoot cultures of apple scion Húsvéti rozmaring after four weeks of culture.

Shoot Length (cm)					
Main effect of different factors					
Cytokinin *** Concentration *** Cytokinin × Concentration interaction ***					
Cytokinins					
TDZ		BA	BAR	*m*T	KIN
0.48 ± 0.01 c		1.56 ± 0.034 a	1.50 ± 0.035 a	1.29 ± 0.07 b	0.01 ± 0.01 d
Concentrations (μM)					
Ø		2	4	6	8
0.15 ± 0.02 d		0.93 ± 0.045 c	1.13 ± 0.043 b	0.96 ± 0.035 c	1.22 ± 0.039 a
Treatments					
μM	TDZ	BA	BAR	*m*T	KIN
Ø	0.15 ± 0.02 fgh				
2	0.51 ± 0.01 e	1.80 ± 0.08 a	1.45 ± 0.09 abc	0.47 ± 0.14 ef	0.02 ± 0.02 h
4	0.35 ± 0.01 efg	1.51 ± 0.06 abc	1.64 ± 0.07 ab	1.46 ± 0.09 abc	0.00 ± 0.00 h
6	0.48 ± 0.01 e	1.24 ± 0.06 c	1.51 ± 0.05 abc	0.91 ± 0.15 d	0.05 ± 0.05 gh
8	0.54 ± 0.01 e	1.74 ± 0.05 ab	1.43 ± 0.06 bc	1.70 ± 0.13 ab	0.00 ± 0.00 h

Values followed by the same letter in the same column are not significantly different at *p* ≤ 0.05 level, according to Tukey’s multiple range test. Ø (CK-free MS medium) served as a control. *** means *p* ≤ 0.001.

## Data Availability

Data are contained within the article.

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
