# Peer review of "Meta-Topolin as an Effective Benzyladenine Derivative to Improve the Multiplication Rate and Quality of In Vitro Axillary Shoots of Húsvéti Rozmaring Apple Scion"

_plants, 2024, doi:10.3390/plants13111568_

Round 1
Reviewer 1 Report
Comments and Suggestions for Authors
The paper is well written, and the authors well described the previous studies available on the effects of the application of metatopoline on multiplication phase and quality of explants in micropropagation in other fruit species, including apple.
However, the manuscript has some very critical points. On my opinion, it does not represent a consistent and effectively innovative contribution in respect to the literature available on this topic on apple. The innovative aspect mainly relays on the shoot multiplication response of a single apple cultivar for one subculture
Adding some more results doing more subcultures, I think, is would be also advisable, since the response of explants to cytokinin cultured for only 1 subculture (4 weeks) tends frequently to change after more sub-cultures.
In addition, the results presented in the tables on the effective positive effect of metatopoline in respect to the other cytokinins used need to be improved in terms of statistical analysis: in fact, I found some discrepancy with the data interpretation in the test. I believe that the interpretation and the discussion of significance of differences of the data presented has to be reconsidered according to the letters of the table resulting from the statistical analysis. I suggest keeping in mind, if I am not wrong, that means like 5.40± 0.321 a, 4.28± 0.286 abcd, 4.92± 0.282 ab etc. are not significantly different. Thus, I suggest adapting congruently the description of the results. I also encourage adding the values of the SE in the data in the table line of the overall cytokinin effect and concentration to give some information on homogeneity of data calculated.
I have also to suggest doing some improvement in the figure: the vessels with the shoots give, on my opinion, a limited information. Thus, I would substitute them with a single shoot for each treatment giving more information on the shoot morphology.
Author Response
Reply to Reviewer 1
The paper is well written, and the authors well described the previous studies available on the effects of the application of meta-topolin on multiplication phase and quality of explants in micropropagation in other fruit species, including apple.
Response: Many thanks for your encouraging comments and evaluation of our work
However, the manuscript has some very critical points. On my opinion, it does not represent a consistent and effectively innovative contribution in respect to the literature available on this topic on apple. The innovative aspect mainly relays on the shoot multiplication response of a single apple cultivar for one subculture.
Adding some more results doing more subcultures, I think, is would be also advisable, since the response of explants to cytokinin cultured for only 1 subculture (4 weeks) tends frequently to change after more sub-cultures.
Response: Yes, we can agree, subculturing on the same cytokinin can cause changes in the multiplication and as an after-effect in the subsequent rooting after more subcultures. The after-effect is typical in the case of TDZ and BA when culturing apple in vitro. However, with these experiments our aim was to select the cytokinin potentially can be used for shoot multiplication of apple cultivar Húsvéti rozmaring. As a results of our current experiment presented in the manuscript, it is clear that the majority of cytokinin types (BA, TDZ and KIN surely) can be excluded as potential cytokinin that can be used for the shoot multiplication of this scion, because their effects were non-beneficial already in the first subculture, the developed shoots were hardly, or not suitable for subculturing. According to the apple in vitro literature cited in the manuscript, as well, TDZ or BA can be replaced by BAR or mT if previous cytokinins cause hard side-effects, but it is highly genotype-dependent. In the case of this scion our current findings show that BAR was not able to overcome the side-effects, only mT was able to do that. In the next phase of our research we intend to study the effects of subcultures, and the after-effects of mT on the subsequent rooting and acclimatization (if there is any).
Thank you again for this comment, and we have added a paragraph regarding the role of subculturing to the manuscript (Discussion section).
In addition, the results presented in the tables on the effective positive effect of metatopolin in respect to the other cytokinins used need to be improved in terms of statistical analysis: in fact, I found some discrepancy with the data interpretation in the test. I believe that the interpretation and the discussion of significance of differences of the data presented has to be reconsidered according to the letters of the table resulting from the statistical analysis. I suggest keeping in mind, if I am not wrong, that means like 5.40± 0.321 a, 4.28± 0.286 abcd, 4.92± 0.282 ab etc. are not significantly different. Thus, I suggest adapting congruently the description of the results. I also encourage adding the values of the SE in the data in the table line of the overall cytokinin effect and concentration to give some information on homogeneity of data calculated.
Response: We revised and rewrite the results according to letters of the statistical analysis of the data. We added the values of the SE of the main effects in the tables.
I have also to suggest doing some improvement in the figure: the vessels with the shoots give, on my opinion, a limited information. Thus, I would substitute them with a single shoot for each treatment giving more information on the shoot morphology.
Response: The typical morphology can be seen well if the whole shoot bunch is presented, mainly in the cases where dwarf growth and hyperhydricity and therefore the barely cutable shoot development occurred. Anyway, we modified the figure according to the available photos with us, we have photos of single shoots for BA, TDZ, and CK-free treatments. However, we have deleted the photos of the jars in cases of BA, TDZ, BAR, mT and CK-free medium, so that the morphology can be seen better also in the case of the removed shoot bunches.
Many thanks again for your comments which aimed to improve the MS. All changes or corrections you asked us to do are highlighted in yellow in the manuscript text.
We trust you can accept our responses and corrections in the manuscript.

Reviewer 2 Report
Comments and Suggestions for Authors
The manuscript is trying to establish optimum concentration of various cytokinins for efficient shoot multiplication of apple scion Húsvéti rozmaring. I have made further comments in the attached pdf.
The abstract has no conclusion. Please add one.
Very little information is given about Húsvéti rozmaring. Please remedy this.

In the “Materials and methods” section the authors keep changing tenses: from “were separated” at the beginning of one sentence, to “have been removed” to “shoots contained” towards the end. Please keep using the same tense throughout the paper.
Frequently in the “Results” section words that should be plural are singular or the other way around. This makes understanding what the authors are trying to say difficult. Please revise the section and correct language mistakes.
Author Response
Reply to Reviewer 2
The manuscript is trying to establish optimum concentration of various cytokinins for efficient shoot multiplication of apple scion Húsvéti rozmaring. I have made further comments in the attached pdf.
Response: Many thanks for your comments which aimed to improve the MS. All changes or corrections you asked us to do are made and highlighted in yellow in the manuscript.
The abstract has no conclusion. Please add one.
Response: A conclusion is added.
Very little information is given about Húsvéti rozmaring. Please remedy this.
Response: the information about Húsvéti rozmaring in English literature is really rare. We could find some information in English in Dobránszki et al. 2000b; and in this cite https://www.fondazioneslowfood.com/en/ark-of-taste-slow-food/husveti-rozmaringapple/ accordingly we inserted the following part in introduction section. We think, it would be better than M and M section.
Húsvéti rozmaring (HR) or Entz rozmaring apple scion; is considered one of the most popular and traditional old apple varieties of gene-preserving and gene-resource importance; which is well known and widespread in Hungary. The exact origin of this variety is not clear, but it may have originated in the Great Hungarian Plain area that lies between the Danube and Tisza rivers, and is known for its sandy soils. It can be used as a source of resistance in breeding programs of apple plant due to its tolerance against different biotic and abiotic stresses. The resistance of this variety makes it a good candidate for organic cultivation. It is diploid, self-fertile and late-ripening cultivar. It shows excellent post-harvest ability therefore its fruits that are usually harvested in October can be stored at 10° C till about Easter time, at the end of March. Its fruits are yellowish-green with a red blush colour at ripening.
In the “Materials and methods” section the authors keep changing tenses: from “were separated” at the beginning of one sentence, to “have been removed” to “shoots contained” towards the end. Please keep using the same tense throughout the paper.
Response: Thank you, according to your comment we have fully text edited the text. We have edited in M and M section accordingly, and checked the tenses in the other parts of the MS; the tense of the sentences depends on the time of occurrence of the verb and its continuity in the present or in the past, passive or active.
Frequently in the “Results” section words that should be plural are singular or the other way around. This makes understanding what the authors are trying to say difficult. Please revise the section and correct language mistakes.
Response: Thank you for this comment, we have fully text edited the manuscript.
Many thanks again for your comments which aimed to improve the MS. We trust you can accept our responses and corrections in the manuscript.

Round 2
Reviewer 1 Report
Comments and Suggestions for Authors
The manuscript has been highly improved taking accurately into consideration the suggestions of the previous revision. I have to say that , initially, I had some doubts concerning the innovative aspects of the contribution because the authors focused the research only on a single variety. However, the high importance of this variety was accurately described by the authors’ in the reviewed version. Consequently, taking into consideration all the aspects improved in the present manuscript version, I think that the manuscript now represents a contribution of satisfactory scientific an technical level
Small few comments
Line 97 --- apple scion is considered one---
Put In vitro in italics everywhere (in some points it is not in this stile) es. in legend of Fig.1 and line 387
Author Response
Reviewer 1_v2
The manuscript has been highly improved taking accurately into consideration the suggestions of the previous revision. I have to say that, initially, I had some doubts concerning the innovative aspects of the contribution because the authors focused the research only on a single variety. However, the high importance of this variety was accurately described by the authors’ in the reviewed version. Consequently, taking into consideration all the aspects improved in the present manuscript version, I think that the manuscript now represents a contribution of satisfactory scientific an technical level
Small few comments
Line 97 --- apple scion is considered one---
Put In vitro in italics everywhere (in some points it is not in this stile) es. in legend of Fig.1 and line 387
Response: Ok, done in all MS. I check all in vitro to be italic, and it is highlighted in green
Many thanks for your encouraging comments and evaluation of our work
The improvement done on the MS is due to you revision

Reviewer 2 Report
Comments and Suggestions for Authors
The manuscript can be published in present form.
Author Response
Reviewer 2_v2
The manuscript can be published in present form.
English language is fine. No issues detected
Response: Many thanks for your effort done to evaluate the MS
